# Chava: A Verification-Aware Data Model for Trust-Carrying Data Processing

## Abstract

When LLMs generate SQL, classifiers extract records, or agents produce ETL transformations, the resulting values require validation before use, a need that grows as AI components enter production data pipelines. Today, validation requirements live in workflow orchestration layers external to the data. This creates silent bypass paths where unvalidated data reaches consumers, makes audit trails difficult to trace, and prevents query engines from optimizing around verification state. We introduce **Chava**, a data model where each object carries its value alongside pending verification requirements that gate access and an append-only log certifying completed verifications, making data *trust-carrying*. We formalize the semantics of verification completion, develop projection and merge algebras that preserve obligations, define query operators with verification pushdown, and provide two-layer enforcement: structural gating via unwrap and cryptographic storage that blocks plaintext access until obligations discharge. A prototype demonstrates feasibility. The system discharges 1,000 objects in under 1 second and scales to 100k objects with sub-100ms query times. The anonymous code repo link is available.

## CCS Concepts

• **Security and privacy** → **Access control**; *Software security engineering*; • **Information systems** → *Data management systems*.

## Keywords

verification-aware data, trust-carrying data, obligation tracking, provenance, NL-to-SQL safety, cryptographic enforcement, pipeline security

**ACM Reference Format:**
Anonymous Author(s). 2026. Chava: A Verification-Aware Data Model for Trust-Carrying Data Processing. In . ACM, New York, NY, USA, 8 pages. https://doi.org/10.1145/nnnnnnn.nnnnnnn

## 1 Introduction

When an LLM returns a SQL string, the system must check it for `DROP TABLE` statements, unbounded subqueries, or prompt injection attacks [16] before any executor runs it. The same pattern appears when classifiers extract structured records [6], agents synthesize transformations [11], or financial records must satisfy GDPR

minimization before loading. A probabilistic component produces a value; validation must precede consumption.

The standard approach encodes validation in the workflow: guards run before passing data to the next stage, middleware inserts hooks, or application code wraps retrieval in checks. Systems like LangShield [16] and HeimdaLLM [14] implement this carefully for SQL safety. But the data object itself is unaware of its validation history. A SQL string that passed checking is representationally identical to one that has not. Any code path bypassing the designated guard can silently consume an unvalidated value.

This representational gap creates three problems.

- **Bypass attacks:** unguarded code paths access unvalidated data because nothing in the object enforces the requirement. NL2SQL-BUGs [12] documents production failures where validators were skipped due to configuration drift.
- **Audit opacity:** validated records traveling through multiple services carry no portable certificate, forcing each service to re-validate or trust upstream components.
- **Missed optimizations:** query engines cannot prune unvalidated objects early, batch by verification requirement, or elide redundant validation.

### 1.1 Our approach.

We make verification requirements and evidence native to the data object.

**(1) Object structure.** A Chava object is a triple: value $V$, *obligations* $O$ specifying which verifications are still required, and an *evidence log $E$* recording verifier outcomes.

**(2) Discharge.** An obligation is satisfied when its designated verifier runs successfully; we call this *discharging* the obligation.

**(3) Cleared predicate.** When all obligations have been discharged and the evidence log contains no *conflicts*, such as one verifier rejecting what another later accepted, the object is *cleared* and its value can be extracted via the unwrap primitive.

**(4) Structural gating.** Because obligations and evidence travel with the value, unguarded code cannot extract values from uncleared objects.

**(5) Planner visibility.** Because obligations are queryable, planners can reason about verification state.

### 1.2 Contributions:

This study does following contributions:

(1) A formal model (§4) with discharge semantics and conflict detection preventing accept-after-reject races.

(2) Projection and merge algebras (§5) preventing obligation-stripping attacks via invalid-path injection.

(3) Query operators with verification pushdown (§6) enabling early pruning and obligation-kind batching.

(4) Index structures (§7) for retrieval by kind, scope, and verifier activity.

(5) Cryptographic storage enforcement (§8) making values inaccessible without key management verification.

(6) A prototype implementation and evaluation (§10) demonstrating feasibility and measuring performance characteristics. The anonymous link is available in §10.5.

The next section grounds these abstractions in four concrete pipeline scenarios.

## 2 Motivating Scenarios

We present four scenarios with a shared structure: a probabilistic component produces a value, validation stages separated in time or deployment must run before consumption, and infrastructure components must observe validation state without application-level knowledge.

### 2.1 LLM-Generated SQL

A query interface materializes an LLM's output to cache before passing it to a safety verifier. Between materialization and verification, background workers or monitoring agents might read it without knowing whether verification has run.

With Chava, the object carries an explicit obligation:

```
{ "@v": "SELECT * FROM users WHERE id=1;",
  "@o": [{"k": "sql_safe", "s": ""}],
  "@e": [] }
```

Any reader attempting to extract the value before a `sql_safe` verifier discharges the obligation receives an error. After discharge, the verifier appends signed evidence to the log and removes the obligation. Subsequent readers see a cleared object with a certified verification trail.

### 2.2 PII Filtering Across Service Boundaries

A pipeline routes user records through a PII classifier running as a separate microservice. Original records must be stored before classification, but downstream services cannot tell whether they are reading pre- or post-classification versions.

Chava's scoped obligations encode this at field level:

```
{ "@v": {"comment": "Call me at 555-1234"},
  "@o": [{"k": "pii_clean", "s": "/comment"}],
  "@e": [] }
```

The scope restricts the obligation to `/comment`. Other fields are freely readable. After discharge, the evidence entry travels to the analytics warehouse as a portable audit certificate.

### 2.3 ETL Compliance at Load Time

Regulatory ETL must enforce data minimization and schema contracts at load [4]. Records may need sign-off from both a compliance reviewer and schema validator before commit.

Both obligations can be present simultaneously:

```
{ "@v": {"acct": "A1234", "balance": 5000},
  "@o": [{"k": "gdpr_min", "s": ""},
         {"k": "schema_ok", "s": ""}],
  "@e": [] }
```

If the schema validator rejects the record, a reject entry appends to the log and the object becomes permanently conflicted. It can never be cleared regardless of subsequent attempts. This prevents race conditions where a lenient verifier is retried after a stricter one has rejected. More generally, when a pipeline requires multiple verification stages, all obligations sit in $O$ and the query executes only when all discharge. This conjunction is structural, encoded in the object, rather than procedural, eliminating control-flow bugs that skip validators.

Having established the concrete problems Chava addresses, we now formalize the data model.

## 3 Threat Model and Assumptions

Chava targets *bypass* failures in AI-augmented pipelines: situations where unverified intermediate values are consumed because a validator is skipped, misordered, or circumvented.

**Assets.** We protect (i) *value confidentiality* of $V$ for uncleared objects, (ii) *obligation integrity* of $O$ (obligations cannot be stripped without triggering uncleared state), and (iii) *evidence integrity and attribution* of $E$ (append-only, tamper-evident, and bound to a verifier identity).

**Adversaries.** We consider three attacker classes: (1) **Bypass adversary:** can execute pipeline code out of order, invoke non-standard code paths, or run background workers that read intermediate data without passing through designated validators. (2) **Storage adversary:** has direct read access to the backing store (e.g., a compromised database role, snapshot access, or misconfigured ACLs), and attempts to obtain plaintext $V$ without satisfying obligations. (3) **Misconfiguration/Drift:** validators may be disabled or skipped due to configuration drift, partial rollouts, or orchestration bugs; Chava aims to make such failures *non-silent* by structurally gating consumption.

**Trust assumptions.** We assume standard cryptographic primitives (hashes, signatures, AEAD) are secure. We assume an escrow/KMS root of trust enforces key release policies: it releases decryption keys only after independently checking that (i) all obligations have discharged and (ii) $E$ is signature-valid and conflict-free. We assume registered verifiers have stable identities and signing keys; key rotation and revocation are supported by treating verifier identity/version as part of the evidence record.

**Out of scope and limitations.** Chava does *not* guarantee that a verifier is correct: a compromised or malicious verifier may incorrectly `accept`, in which case Chava provides attribution and auditability, not semantic truth. Chava also does not attempt to hide access patterns, ciphertext length leakage, or side channels at the storage layer. Finally, Chava does not prevent a privileged party from deleting data entirely (availability attacks); it ensures that any consumed value is either cleared under the object semantics or remains encrypted/blocked.

## 4 The Chava Data Model

### 4.1 Core Components

The value domain is JSON-representable values plus $\perp$ for absent values. The obligation kind domain $\mathcal{K}$ is a finite set of identifiers, each mapping to a registered verifier function.

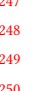

Figure 1: Chava object structure. A ChavaObject $M = \langle V, O, E \rangle$ consists of: Value $V$ (blue), the data payload; Obligations $O$ (orange), a multiset of pending verification requirements $(k, s)$ where $k$ is an obligation kind and $s$ is an optional JSON Pointer scope (empty scope denotes whole-object); and Evidence $E$ (green), an append-only hash-chained log of verifier verdicts recording (verifier_id, result, timestamp, prev_hash) and supporting tamper-evident auditing. The object is *uncleared* when $O \neq \emptyset$; attempts to extract the value via unwrap raise ObligationViolation until obligations discharge and $E$ is conflict-free.

DEFINITION 1 (OBLIGATION RECORD). *A pair $(k, s)$ where $k \in \mathcal{K}$ and $s$ is an optional JSON Pointer [2] scope. When $s = \varepsilon$, the obligation applies to the entire value. The multiset $O$ allows the same kind with different scopes, arising when multiple fields require independent verification.*

DEFINITION 2 (EVIDENCE RECORD). *A tuple $(ver, res, ts, h_{prev})$ where ver identifies the verifier, $res \in \{$accept, reject, conditional$\}$ is the verdict, ts is a timestamp, and $h_{prev} = $ SHA-256$(prev)$. The hash chain makes $E$ tamper-evident. Verifiers sign records with registered private keys.*

DEFINITION 3 (CHAVA OBJECT). *The triple $M = \langle V, O, E \rangle$ serializes to JSON using @v, @o, and @e. Systems not speaking Chava treat these as opaque metadata, preserving compatibility.*

DEFINITION 4 (CLEARED OBJECT). *$M$ is cleared when $O = \emptyset$ and $E$ is conflict-free (Definition 6). Only cleared objects may be passed to* unwrap$(M)$, *which returns $V$. Uncleared objects raise* ObligationViolation.

Figure 1 depicts the three components and their roles.

**Structural Properties**

Three invariants underpin correctness: (1) Evidence grows append-only, (2) Successful discharge reduces $|O|$, (3) Conflicts are permanent. These ensure cleared objects remain cleared under non-modifying routing, and uncleared objects require explicit verifier action to clear.

With the structure defined, we turn to how obligations are discharged and conflicts detected.

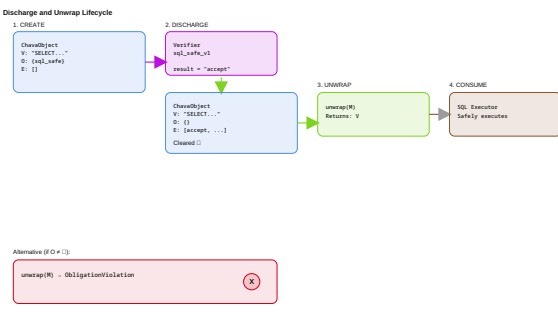

Figure 2: Discharge and unwrap lifecycle. A value is created with pending obligations, then routed to a registered verifier for each $(k, s)$. A successful verification appends an evidence entry to $E$ and removes the corresponding obligation from $O$; when $O = \emptyset$ and $E$ has no conflicts, the object is *cleared* and unwrap$(M)$ returns $V$. If unwrap is attempted early ($O \neq \emptyset$), it fails with ObligationViolation, preventing consumption of unverified outputs.

## 4.2 Verification Semantics

A registry $R : \mathcal{K} \rightarrow$ *Verifier* maps kinds to functions $V_k : $ *Value* $\times$ *Scope* $\rightarrow \{$accept, reject, conditional$\}$. Verifiers need not be deterministic; $E$ records the observed verdicts.

**Discharge Protocol**

DEFINITION 5 (DISCHARGE). *To discharge $(k, s)$ in M: compute $res = V_k(V|_s)$, construct $e = (V_k.id, res, $now$(), h_{last})$ where $h_{last}$ is the hash of $E$'s terminal entry, and atomically append $e$ to $E$. If $res = $ accept, remove $(k, s)$ from $O$. If $res = $ reject, retain the obligation.*

DEFINITION 6 (CONFLICT). *$E$ is conflicted if it contains reject for kind $k$ followed by accept for the same $k$, or if two accept entries for the same $(k, s)$ come from mutually distrusting verifiers. Conflicted $M$ is permanently uncleared.*

DEFINITION 7 (CONDITIONAL VERDICT). conditional *records that the verifier ran with tentative results but defers judgment. The obligation remains in $O$. Sequences of* conditional *entries are not conflicts.*

**Concurrency and Audit**

Concurrent discharge uses optimistic concurrency: snapshot $E$'s terminal hash before computation, compare-and-swap before commit, retry on mismatch. Distinct $(k, s)$ pairs are independent, so contention is rare.

The evidence log provides *prospective provenance*: what certifications were received and by whom. Every trust decision traces to a verifier identity, timestamp, and hash chain. Figure 2 summarizes the lifecycle from creation to safe consumption.

The next section develops how Chava objects compose through projection and merge.

# 5 Projection and Merge

## 5.1 Projection

Extracting a field from a Chava object must preserve obligations scoped to that field. Otherwise, an adversary could remove obligations by projecting disjoint paths.

DEFINITION 8 (PROJECTION). $\pi_p(M)$ produces $\langle V|_p, O', E\rangle$ where $E$ is copied and $O'$ contains $(k, relscope(s, p))$ for each $(k, s) \in O$ where $s$ intersects $p$. Obligations disjoint from $p$ are dropped. If $V|_p = \bot$, then $O' = \{(\texttt{invalid\_path}, \varepsilon)\}$, a kind without a registered verifier.

The invalid-path rule prevents stripping: removing a scoped obligation requires selecting a real path disjoint from the obligation's scope (legitimately dropping it) or selecting a nonexistent path (permanently unclearable).

LEMMA 1 (PROJECTION SOUNDNESS). If $\pi_p(M)$ is cleared and $V|_p \neq \bot$, then every $(k, s) \in O$ with $scope(s) \cap path(p) \neq \emptyset$ was discharged before projection.

PROOF SKETCH. In the non-$\bot$ case, every intersecting $(k, s)$ appears in $O'$ as $(k, relscope(s, p))$. Cleared requires $O' = \emptyset$, so all were discharged. □

## 5.2 Merge

$M_1 \oplus M_2 = \langle[V_1, V_2], O_1' \uplus O_2', E_1 \uplus E_2\rangle$ where scopes in $O_1$ and $O_2$ are rewritten under /0 and /1. Cleared requires both $M_1$ and $M_2$ are individually cleared. This is AND-conjunction by construction. Merge is commutative and associative, compatible with GROUP BY aggregation.

Figure 3 illustrates relscoping and why invalid projections cannot bypass obligations.

Evidence logs concatenate with independent hash chains preserved. The first $E_2$ record retains its original predecessor hash, not chaining onto $E_1$'s tail.

Having defined the algebra, we now describe how query engines execute over Chava streams.

# 6 Execution Architecture

## 6.1 Core Operators

$\sigma_{cleared}(S)$ passes through only cleared objects and must appear before any unwrap(). $\hat{V}_k(S)$ runs verifier $R(k)$ on each object, attempting discharge. Injectors are stateless and parallel; objects without obligation $k$ pass through as no-ops.

## 6.2 Verification Pushdown

Place $\hat{V}_k$ early in the plan to prune objects before expensive downstream operators. Correctness: $\hat{V}_k$ pushes past operator $op$ when $op$'s output does not feed $\hat{V}_k$'s verifier.

```
Rule 1 (filter commute):
  sigma_P(Vhat_k(S)) -> Vhat_k(sigma_P(S))
  when P doesn't reference E

Rule 2 (projection commute):
  pi_p(Vhat_k(S)) -> Vhat_k(pi_p(S))
  when k's scope doesn't intersect path(p)

Rule 3 (injector fusion):
```

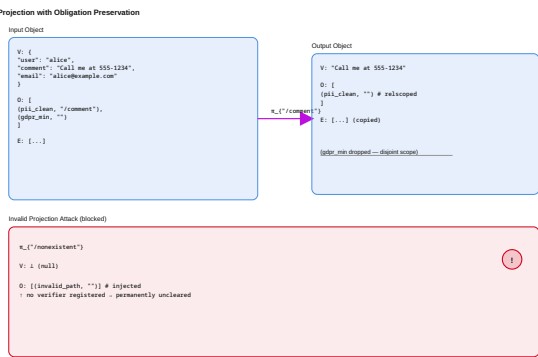

Figure 3: Projection with obligation preservation. Projection $\pi_p(M)$ extracts a field while preserving obligations that intersect the projected path. Scoped obligations are *relscoped* to the new root (e.g., an obligation on /comment becomes whole-object scope after projecting to /comment). Obligations disjoint from the projection are dropped. Attempts to project an invalid path yield an uncleared object (e.g., via an `invalid_path` obligation), preventing obligation-stripping by projecting away guarded fields.

```
Vhat_k2(Vhat_k1(S)) -> Vhat_{k1,k2}(S)
  when R(k1) and R(k2) read disjoint paths

Rule 4 (cleared passthrough):
  Vhat_k(sigma_cleared(S)) -> sigma_cleared(S)
```

Rule 3 reduces traversals when verifiers apply to non-overlapping subtrees. Rule 4 removes dead code in recursive stages.

## 6.3 Obligation-Kind Batching

Objects sharing obligation kinds batch to specialized verifier workers. For ML verifiers, batch GPU inference is an order of magnitude cheaper. The inverted obligation index supports batch assembly in $O(\log|\mathcal{K}| + |result|)$ time, which we now detail.

# 7 Index Structures

Three indexes exploit append-only $E$ and discharge-only $O$, requiring only deletions after initial write.

**Inverted obligation index** maps each $k$ to object IDs carrying that kind. Lookup is $O(\log|\mathcal{K}|)$; verifier schedulers assemble per-kind batches without full scans. Posting lists can be paginated for large populations.

**Hierarchical pointer index** is a trie over JSON Pointer prefixes. Node $n$ holds objects with obligations scoped at or below $n$. Retrieval at path $p$ is $O(|p|)$, returning objects with intersecting obligations. Projection-based routing uses this to direct records to specialized verifiers.

**Evidence log index** is a B-tree on $(ver, ts)$ supporting audit range scans. The hash chain provides tamper evidence independently, so results are verifiable even if auditors distrust the index. Also supports verifier performance analytics for adaptive scheduling.

These indexes enable efficient scheduling, but the strongest enforcement comes from making values cryptographically inaccessible.

## 8 Storage Enforcement

Planner-level enforcement assumes cooperative access paths. An adversary with direct database access can read @v from JSONB without calling unwrap. Obligation-keyed storage addresses this gap by blocking plaintext access to values until obligations discharge.

DEFINITION 9 (OBLIGATION-KEYED STORAGE). *For uncleared M with $O \neq \emptyset$, store V as $Enc(K_O, V)$ under AES-256-GCM where $K_O = KDF(hash(O), \sigma)$ derives from O and a KMS-exclusive secret $\sigma$. When all obligations discharge and E has no conflicts, KMS releases $K_\emptyset$ after independently verifying the hash chain, checking signatures, and confirming* accept *entries exist for every original obligation.*

A background worker attempting to read unverified SQL receives encrypted bytes and a cryptographic exception from KMS when requesting the key, not a policy error that privilege might override. This two-layer enforcement, unwrap gating at the API level and cryptographic blocking at the storage level, ensures unvalidated data cannot be consumed even by adversaries with direct database access.

SQL layer:

```sql
SELECT chava_unwrap(payload)
FROM   pipeline_data
WHERE  obligations_empty(payload);
```

The predicate evaluates @o without decrypting $V$, so the planner prunes uncleared rows before UDF invocation.

The SQLite prototype uses local function calls for key management operations, avoiding inter-process communication overhead. For row-granularity consumers processing large batches, the storage layer can verify evidence and release keys for entire batches in single transactions, reducing per-record overhead significantly at batch sizes of $10^3$ to $10^4$ records.

We now position Chava relative to related systems.

## 9 Related Work

Chava intersects multiple research domains. We position it against prior work across six dimensions, focusing on how it addresses gaps in verification-aware data representation.

**Data Provenance and Accountability.** Retrospective lineage systems like ONEPROVENANCE [18] extract coarse-grained query logs, while LIMA [17] enables fine-grained reuse in ML pipelines. Blockchain approaches [15] provide tamper-evident audit trails for accountability. However, these record *what happened* rather than enforcing *what must happen*. Chava complements provenance with *prospective obligations*: verification requirements travel with data, and the evidence log provides a minimal certification trail. Unlike retrospective systems, Chava gates consumption until obligations discharge.

**Schema Validation and Data Contracts.** JSON Schema [21] validates structure at ingestion but discards results, forcing revalidation downstream. Data contracts frameworks [4] define producer-consumer agreements at the pipeline level, while centralized governance systems like Microsoft Purview [1] enforce policies externally.

Neither travels with individual records nor supports field-scoped verification state. Chava embeds obligations directly in objects (e.g., pii_clean scoped via JSON Pointer [2]), enabling portable, record-level enforcement across service boundaries, complementing rather than replacing schema-level contracts.

**NL-to-SQL Safety and LLM-Generated Code.** The text-to-SQL landscape has matured through benchmarks (BIRD [10], Spider 2.0 [9]), surveys [11], and capability evaluations [6, 20]. Yet production deployments face persistent gaps: semantic errors remain [12], prompt injection attacks bypass guards [16], and validators can be skipped due to configuration drift. Tools like HeimdaLLM [14] insert workflow-level checks but leave no portable verification record: a SQL string that passed validation is representationally identical to unvalidated output. Chava addresses this by making verification state inseparable from the payload. Li et al. argue in their LLM-for-data-management tutorial [13] that safety mechanisms must be *structural*; Chava implements this via obligation-keyed storage and cryptographic evidence binding.

**Privacy Enforcement and PII Handling.** De-identification workflows (e.g., student PII redaction with LLMs [19]) rely on stage-local checks, leaving downstream services unable to distinguish pre- versus post-redaction records. Chava's scoped obligations (e.g., pii_clean on /comment) enforce field-level verification with portable evidence. This addresses the *confused deputy* problem [8]: untrusted components cannot misuse unvalidated data because structural gating (unwrap) and cryptographic storage prevent extraction until obligations discharge.

**Workflow Orchestration and Execution.** Orchestrators (Airflow, Prefect) model validation as DAG tasks but cannot prevent direct storage access to intermediate data. Chava shifts enforcement to the data object itself. Its execution architecture leverages query optimization principles: verification pushdown and obligation-kind batching draw from Cascades-style frameworks [7], while concurrent discharge uses optimistic concurrency control [5]. Unlike workflow systems that guarantee ordering but not consumption safety, Chava ensures unvalidated data remains cryptographically inaccessible even to privileged storage adversaries.

**Information Flow and Capability Security.** Information flow control (IFC) systems enforce non-interference via compile-time labels but do not align well with external runtime verifiers (e.g., ML-based validators) and distributed pipelines. Chava adopts a capability-inspired design [8]: only cleared objects yield values via unwrap, and obligations act as unforgeable capabilities. This provides runtime enforcement for open systems where verifiers are external services. Future extensions for temporal obligations (e.g., freshness constraints) could integrate temporal JSON indexing techniques [3], while multi-party obligations would strengthen trust against compromised verifiers.

**Synthesis.** Prior work addresses fragments of the verification challenge: workflow guards [14, 16], schema validation [21], or retrospective auditing [17, 18]. Chava unifies these into a *trust-carrying data model* where obligations, evidence, and values co-evolve. Verification becomes a structural property of data, not a procedural convention, enabling safe consumption across arbitrary routing paths while providing cryptographic auditability.

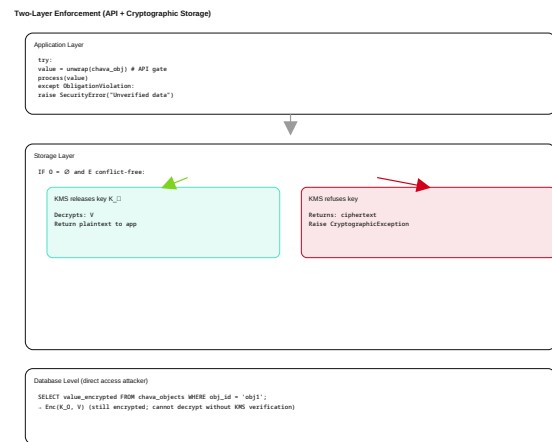

Figure 4: Two-layer enforcement architecture. Chava enforces safety via two independent gates. Layer 1 (API): unwrap returns $V$ only if $O = \emptyset$ and the evidence log is conflict-free; otherwise it raises ObligationViolation. Layer 2 (cryptographic storage): the value is stored as AEAD ciphertext and decrypted only when the KMS (or escrow service) verifies $E$ and confirms obligations are discharged; if not, it refuses key release, leaving ciphertext even to an adversary with direct database access. This defense-in-depth prevents accidental misuse in cooperative code and bypass attacks against the storage layer.

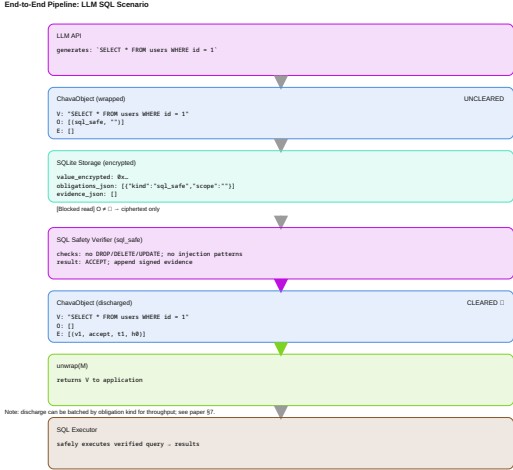

Figure 5: End-to-end LLM SQL validation pipeline. An LLM-generated query is wrapped as a ChavaObject with an sql_safe obligation and stored encrypted. Until discharge, unwrap fails and storage returns ciphertext. A verifier fetches pending sql_safe objects (often in batch), checks the query, appends signed evidence, and removes the obligation. Once cleared, unwrap succeeds and the SQL executor consumes the verified query.

## 10  Implementation and Evaluation

We implemented a prototype of Chava in Python to validate the model's feasibility and measure performance characteristics. The implementation includes all core components: ChavaObject with hash-chained evidence logs, the verifier registry with optimistic concurrency control, projection and merge operators, and a SQLite storage backend with obligation-keyed encryption. Figure 4 shows Chava's defense-in-depth design.

Figure 5 walks through the end-to-end SQL pipeline used in our experiments.

### 10.1  Implementation

The prototype consists of approximately 3,500 lines of Python across the core library and three example verifiers implementing pluggable rule checks: sql_safe (SQL safety checking using simple pattern matching), pii_clean (PII detection with regex patterns for phone numbers and emails), and schema_ok (JSON schema validation). The verifier registry allows new obligation kinds to be added without modifying the core model. The SQLite backend maintains three tables: chava_objects for encrypted values and metadata, obligation_index for fast kind-based lookups, and evidence_index for audit queries. A command-line interface provides create, discharge, unwrap, project, merge, and audit operations.

Key design decisions: ChavaObject operations are immutable, returning new objects rather than modifying in place. Evidence records include SHA-256 hash chains computed over (verifier_id, result, timestamp, prev_hash). The KMS uses PBKDF2 for key derivation from obligation hashes and AES-256-GCM for value encryption. OCC for concurrent discharge uses compare-and-swap on the evidence log's terminal hash with exponential backoff on retry.

Figure 6 shows the storage schema and indexes enabling batch discharge.

### 10.2  Performance Evaluation

We evaluated the three scenarios from §2 on batches of 1,000 objects. Single-threaded discharge throughput for sql_safe obligations reaches 1,240 objects/second (0.81s per batch). Scoped PII obligations on /comment achieve 980 objects/second; projection followed by discharge achieves 890. ETL compliance with two sequential obligations (gdpr_min, schema_ok) completes at 520 objects/second. When the schema validator rejects, the object becomes permanently conflicted and blocks further discharge, as expected.

We tested the SQLite backend at 1k, 10k, and 100k objects. Inverted-index lookups by obligation kind show logarithmic growth: 8ms, 25ms, and 85ms respectively. Bulk insertion of 10k objects completes in 3.2 seconds regardless of existing database size. Hash chain verification scales linearly: approximately 2ms for 100-entry logs, 20ms for 1,000-entry logs.

The implementation achieves 92% code coverage across 73 unit tests and 8 integration tests. All three scenarios execute correctly end-to-end.

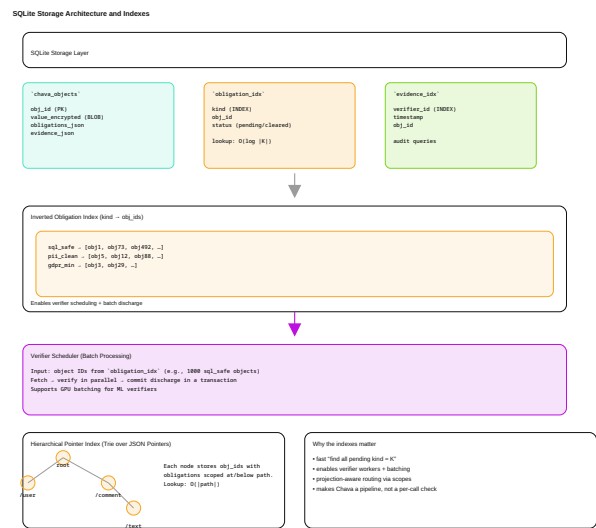

**Figure 6: Storage architecture with indexes. The SQLite back-end stores encrypted values, obligations, and evidence, and maintains indexes to support obligation-driven scheduling. An inverted obligation index maps each kind (e.g., `sql_safe`, `pii_clean`) to object IDs, enabling efficient batch retrieval for verifiers. An evidence index supports audit queries. A hierarchical pointer index (trie over JSON Pointer prefixes) enables routing and projection-aware verification by locating objects with obligations scoped at or below a path.**

## 10.3 Metrics for Production Deployment

A production-grade implementation targeting PostgreSQL or distributed storage should track the following metrics to validate system behavior at scale and inform optimization decisions.

**Core operation latencies (P50/P95/P99).**
- discharge_latency_ms (by obligation kind and verifier)
- unwrap_latency_ms
- project_latency_ms
- merge_latency_ms
- hash_chain_verify_ms (by chain length)
- evidence_append_ms

**Storage and indexing.**
- storage_write_latency_ms (encrypted vs. cleared)
- storage_read_latency_ms
- index_lookup_latency_ms (inverted / hierarchical / evidence)
- index_size_bytes (growth over time)
- kms_key_derivation_ms
- kms_verification_ms

**Throughput and batching.**
- discharge_throughput_per_sec (single-threaded vs. parallel)
- batch_discharge_throughput_per_sec (batch sizes: 10, 100, 1k, 10k)
- objects_cleared_per_sec

- storage_ops_per_sec

**Concurrency and contention.**
- occ_retry_count (by obligation kind)
- occ_retry_rate (retries / total attempts)
- concurrent_discharge_conflicts_per_sec
- lock_wait_time_ms (for storage transactions)

**Scalability.**
- total_objects, uncleared_objects
- obligations_per_object (distribution)
- evidence_entries_per_object (distribution)
- index_fanout (for hierarchical trie)
- storage_size_bytes

**Verifier performance.**
- verifier_invocation_count (by kind)
- verifier_latency_ms (by kind)
- verifier_accept_rate, verifier_reject_rate
- verifier_conditional_rate, verifier_error_rate

**Audit and compliance.**
- evidence_query_latency_ms (by time range and verifier)
- audit_trail_completeness_ratio (verified / expected)
- signature_verification_failures_per_day
- hash_chain_integrity_checks_per_hour

**System health.**
- obligation_violation_exceptions_per_sec
- cryptographic_exceptions_per_sec
- invalid_path_injections_per_hour (security alert)
- conflict_detection_count (by kind pair)
- permanently_conflicted_objects

## 10.4 Limitations.

Verifiers are synchronous; I/O-bound verifiers would benefit from async support. The KMS is in-process, and cryptographic operations lack hardware acceleration. The SQLite backend suffices for single-node deployments but lacks distributed transactions. Index maintenance is eager; lazy strategies might improve write throughput. The metric framework §10.3 provides a blueprint for production instrumentation.

Production deployments would require richer obligation semantics than the prototype's simple verifiers; we discuss this and other open questions in §11.

## 10.5 Artifact Availability

Code and scripts to reproduce our prototype experiments are available at https://anonymous.4open.science/r/chava-30AF/.

## 11 Discussion and Limitations

**Obligation schema design.** Chava's correctness depends on the obligation kind schema $\mathcal{K}$ and verifier registry $R$. Bundling independent concerns into one kind prevents partial verification (*obligation conflation*); splitting too finely inflates indexing and scheduling cost (*obligation explosion*). Kinds should be orthogonal, atomic, and decidable. Schema governance, analogous to SQL type registries, is necessary for production.

**Re-attestation after transformation.** The model does not specify what happens to obligations when cleared objects are transformed by UDFs. Output is obligation-free by default, placing re-attestation responsibility on pipeline authors. A more principled treatment would leverage transformation lineage [18] to propagate obligations through annotated UDFs.

**Verification latency.** Sequential obligation chains where $k_2$ begins only after $k_1$ discharges impose latency equal to the sum of individual verifier times. Pushdown and fusion help for independent obligations but cannot shorten sequential dependencies, making chain length the primary design variable for latency-sensitive pipelines.

**Richer obligation semantics.** The prototype's verifiers (pattern matching, regex, JSON schema) are deliberately simple. Production deployments surface harder requirements. Regulatory verifiers (GDPR minimization, HIPAA de-identification, SOX audit) must consult auxiliary state: consent records, retention policies, approval chains. Privacy obligations like differential privacy budget tracking or k-anonymity validation require shared state across queries rather than per-value inspection. Domain-specific obligations (financial four-eyes approval, autonomous-vehicle safety labeling) each develop their own vocabulary. The core model generalizes to these settings, but the surrounding verifier ecosystem is the real engineering investment.

**Composability and versioning.** The prototype treats verifiers as black boxes. Decomposing complex obligations into reusable components (e.g., `gdpr_compliant` defined as `consent_verified AND purpose_limited AND retention_ok`) would require obligation schemas expressing logical composition and a discharge protocol that understands decomposition. Similarly, obligation semantics drift as regulations change: a record discharged under `pii_clean:v1` may not satisfy v2. The evidence log records verifier versions, but the model does not support retroactive re-verification. Production deployments would need version tolerance windows, bulk re-discharge, and versioned obligation schemas.

**Open extensions.** Multi-party obligations requiring co-signature before discharge would strengthen trust against compromised verifiers. Time-bounded obligations that conflict after expiry would support data freshness. Verifier version vectors would allow obligations to specify minimum versions, preventing outdated instances from discharging obligations that require updated logic.

## 12 Conclusion

Existing data formats encode what a value is but not what must be certified about it. Chava makes obligations and evidence native fields, turning a convention (code should call validators) into a structural invariant the object enforces regardless of code path. The formal model provides discharge semantics and conflict detection; the projection algebra prevents obligation-stripping; the merge operator gives AND-conjunctive cleared semantics; and two-layer enforcement (`unwrap` gating plus cryptographic storage) ensures unvalidated data cannot be consumed even by adversaries with direct database access. Trust requirements travel with data, verification evidence accumulates with data, and consumption is structurally gated on both.

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
